# *Right Time to Learn*: Promoting Generalization via Bio-inspired Spacing Effect in Knowledge Distillation

Guanglong Sun [* 1 2 3]   Hongwei Yan [* 1 2 3]   Liyuan Wang [4]   Qian Li [1]   Bo Lei [3]   Yi Zhong [1 2]

## Abstract

Knowledge distillation (KD) is a powerful strategy for training deep neural networks (DNNs). Although it was originally proposed to train a more compact "student" model from a large "teacher" model, many recent efforts have focused on adapting it to promote generalization of the model itself, such as online KD and self KD. Here, we propose an accessible and compatible strategy named Spaced KD to improve the effectiveness of both online KD and self KD, in which the student model distills knowledge from a teacher model trained with a space interval ahead. This strategy is inspired by a prominent theory named *spacing effect* in biological learning and memory, positing that appropriate intervals between learning trials can significantly enhance learning performance. With both theoretical and empirical analyses, we demonstrate that the benefits of the proposed Spaced KD stem from convergence to a flatter loss landscape during stochastic gradient descent (SGD). We perform extensive experiments to validate the effectiveness of Spaced KD in improving the learning performance of DNNs (e.g., the performance gain is up to 2.31% and 3.34% on Tiny-ImageNet over online KD and self KD, respectively). Our codes have been released on github https://github.com/SunGL001/Spaced-KD.

## 1. Introduction

Knowledge distillation (KD) is a powerful technique to transfer knowledge between deep neural networks

---

[*]Equal contribution  [1]School of Life Sciences, IDG/McGovern Institute for Brain Research, Tsinghua University, Beijing, China [2]Tsinghua–Peking Joint Center for Life Sciences [3]Beijing Academy of Artificial Intelligence, Beijing, China [4]Dept. of Comp. Sci. & Tech., Institute for AI, BNRist Center, Beijing, China. Correspondence to: Liyuan Wang <wly2023@tsinghua.edu.cn>.

*Proceedings of the 42nd International Conference on Machine Learning*, Vancouver, Canada. PMLR 267, 2025. Copyright 2025 by the author(s).

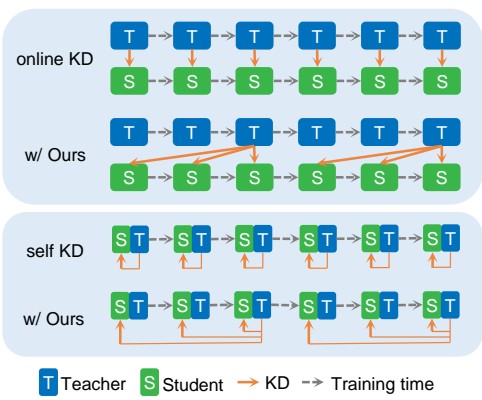

*Figure 1.* **Diagram of Spaced KD.** In online KD, the teacher and student are two individual networks. In self KD, we follow the prior work (Zhang et al., 2019) that distills knowledge from the deepest layer to the shallower layers of the same network. In Spaced KD, we train the teacher with a controllable space interval steps ahead and then distill its knowledge to the student network.

(DNNs) (Gou et al., 2021; Wang & Yoon, 2021). Despite its extensive applications to construct a more compact "student" model from a converged large "teacher" model (aka offline KD), there have been many recent efforts using KD to promote generalization of the model itself, such as online KD (Zhang et al., 2018; Zhu et al., 2018; Chen et al., 2020) and self KD (Zhang et al., 2019; Mobahi et al., 2020). Specifically, online KD simplifies the KD process by training the teacher and the student simultaneously, while self KD involves using the same network as both teacher and student. However, as these paradigms can only moderately improve learning performance, how to design a more desirable KD paradigm in terms of generalization remains an open question.

Compared to DNNs, biological neural networks (BNNs) are advantageous in learning and generalization with specialized adaptation mechanisms and effective learning procedures. In particular, it is commonly recognized that extending the interval between individual learning events can considerably enhance the learning performance, known as the *spacing effect* (Ebbinghaus, 2013; Smolen et al., 2016). This highlights the benefits of spaced study sessions for improving the efficiency of learning compared to continuous sessions, and has been described across a wide range of species from in-

vertebrates to humans (Beck et al., 2000; Pagani et al., 2009; Menzel et al., 2001; Anderson et al., 2008; Bello-Medina et al., 2013; Medin, 1974; Robbins & Bush, 1973).Taking human learning as an example, the spacing effect could enhance skill and motor learning (Donovan & Radosevich, 1999; Shea et al., 2000), classroom education (Gluckman et al., 2014; Roediger & Byrne, 2008; Sobel et al., 2011), and the generalization of conceptual knowledge in children (Vlach, 2014).

Inspired by biological learning, we propose to incorporate such spacing effect into KD (referred to as Spaced KD, see Fig. 1) as a general strategy to promote the generalization of DNNs (see Fig. 2). We first provide an in-depth theoretical analysis of the potential benefits of Spaced KD. Compared to regular KD strategies, the proposed Spaced KD helps DNNs find a flat minima during stochastic gradient descent (SGD) (Sutskever et al., 2013), which has proven to be closely related to generalization. We then perform extensive experiments to demonstrate the effectiveness of Spaced KD, across various benchmark datasets and network architectures. The proposed Spaced KD achieves strong performance gains (e.g., up to 2.31% and 3.34% on Tiny-ImageNet over regular KD methods of online KD and self KD, respectively) without additional training costs. We further demonstrate the robustness of the space interval, the critical period of the spacing effect, and its plug-in nature to a broad range of advanced KD methods.

Our contributions can be summarized as follows: (1) We draw inspirations from the paradigm of biological learning and propose to incorporate its spacing effect to improve online KD and self KD; (2) We theoretically analyze the potential benefits of the proposed spacing effect in terms of generalization, connecting it with the flatness of loss landscape; and (3) We conduct extensive experiments to demonstrate the effectiveness and generality of the proposed spacing effect across a variety of benchmark datasets, network architectures, and baseline methods.

## 2. Related Work

**Knowledge Distillation (KD).** Representative avenues of KD can be generally classified into offline KD, online KD, and self KD, based on whether the teacher model is pretrained and remains unchanged during the training process. Offline KD involves a one-way knowledge transfer in a two-phase training procedure. It primarily focuses on optimizing various aspects of knowledge transfer, such as designing the knowledge itself (Hinton et al., 2015; Adriana et al., 2015), and refining loss functions for feature matching or distribution alignment (Huang & Wang, 2017; Asif et al., 2019; Mirzadeh et al., 2020b). In contrast, online KD simplifies the KD process by training both teacher and student simultaneously and often outperforms offline KD. For instance,

DML (Zhang et al., 2018) implements bidirectional distillation between peer networks. For self KD, the same network is used as both teacher and student (Zhang et al., 2019; Das & Sanghavi, 2023; Mobahi et al., 2020; Zhang & Sabuncu, 2020; Yang et al., 2019; Lee et al., 2019). In this paper, the self KD we refer to is the distillation between different layers within the same network (Zhang et al., 2019; Yan et al., 2024; Zhai et al., 2019). However, existing methods for online KD and self KD often fail to effectively utilize high-capacity teachers over time, making it an intriguing topic to further explore the relationships between teacher and student models in these environments.

**Adaptive Distillation.** Recent studies have found that the difference in model capacity between a much larger teacher network and a much smaller student network can limit distillation gains (Liu et al., 2020a; Cho & Hariharan, 2019; Liu et al., 2020b). Current efforts to address this gap fall into two main categories: training paradigms (Gao et al., 2018) and architectural adaptation (Kang et al., 2020; Gu & Tresp, 2020). For instance, ESKD (Cho & Hariharan, 2019) suggests stopping the training of the teacher early, while ATKD (Mirzadeh et al., 2020a) employs a medium-sized teacher assistant for sequential distillation. SHAKE (Li & Jin, 2022) introduces a shadow head as a proxy teacher for bidirectional distillation with students. However, existing methods usually implement adaptive distillation by adjusting teacher-student architecture from a spatial level. In contrast, Spaced KD provides an architecture- and algorithm-agnostic way to improve KD from a temporal level.

**Flatness of Loss Landscape.** The loss landscape around a parameterized solution has attracted great research attention (Keskar et al., 2016; Hochreiter & Schmidhuber, 1994; Izmailov et al., 2018; Dinh et al., 2017; He et al., 2019). A prevailing hypothesis posits that the flatness of minima following network convergence significantly influences its generalization capabilities (Keskar et al., 2016). In general, a flatter minima is associated with a lower generalization error, which provides greater resilience against perturbations along the loss landscape. This hypothesis has been empirically validated by studies such as (He et al., 2019). Advanced advancements have leveraged KD techniques to boost model generalization (Zhang et al., 2018; Zhao et al., 2023; Zhang et al., 2019). Despite these remarkable advances, it remains a challenging endeavor to fully understand the impact of KD on generalization, especially in assessing the quality of knowledge transfer and the efficacy of teacher-student architectures.

## 3. Preliminaries

In this section, we first present the problem setup and some necessary preliminaries of KD. Then we describe the spac-

ing effect in biological learning and discuss how it may inspire the design of KD.

## 3.1. Problem Setup

We describe the problem setup with supervised learning of classification tasks as an example. Given $N$ training samples $\mathcal{D}_{\text{train}} = \{(x_i, y_i)\}_{i=1}^N$ where $x_i \in \mathbb{R}^d$ and $y_i \in \mathbb{R}^c$, the neural network model $f_\theta(\cdot) : \mathbb{R}^d \mapsto \mathbb{R}^c$ with parameters $\theta \in \mathbb{R}^p$ is optimized by minimizing the empirical risk over $\mathcal{D}_{\text{train}}$ and evaluated over the test dataset $\mathcal{D}_{\text{test}}$. Using the SGD optimizer (Sutskever et al., 2013), $f_\theta(\cdot)$ is updated for each mini-batch of training data $\mathcal{B}_t = \{(x_i, y_i) \in \mathcal{D}_{\text{train}}\}_{i \in \mathcal{I}_t}$, $\mathcal{I}_t \subseteq \{1, 2, \cdots N\}$:

$$\theta_{t+1} = \theta_t - \frac{\eta}{B} \sum_{i \in \mathcal{I}_t} \nabla_\theta L_i(\theta_t), \tag{1}$$

where $L_i(\theta) = l_{\text{task}}(f_\theta(x_i), y_i)$ is a task-specific supervision loss. $\eta$ and $B = |\mathcal{I}_t|$ denote the learning rate and batch size, respectively. KD supports various kinds of interaction between multiple neural networks. The teacher-student framework we refer to here consists by default of a teacher network $g_\phi(\cdot)$ and a student network $f_\theta(\cdot)$, where the flow of knowledge transfer is often one-direction: the learning of $f$ is guided by the output of $g$, but not vice versa. The loss of student network $f$ in KD is bi-component as a weighted sum of task-specific and distillation loss ($l_{\text{task}}$ and $l_{\text{KD}}$), where a hyperparameter $\alpha$ controls the impact of teacher guidance:

$$L_i^{(\text{KD})}(\theta, \phi) = (1 - \alpha)l_{\text{task}}(f_\theta(x_i), y_i) + \alpha l_{\text{KD}}(f_\theta(x_i), g_\phi(x_i)). \tag{2}$$

In many applications, the teacher network $g$ is often different from and much larger than the student network to obtain a more compact model. Meanwhile, there is an increasing number of efforts to implement KD to improve generalization for one particular architecture, where the teacher and student may share a common framework but differ in the random seeds for initialization. Some KD methods even treat different parts within one single network as teacher and student. Below we describe two representative methods:

**Online KD.** Though traditional KD assumes the teacher network $g$ as a pre-trained and powerful model, there exist scenarios where obtaining such a teacher is costly or impractical. Online KD is proposed to learn from an on-the-fly teacher network, allowing for dynamic adaptation during student training. In online KD, the updating of $g$ is aligned with $f$ for every mini-batch $\mathcal{B}_t$ with $\mathcal{I}_t$ (see Alg. 1 in Appendix A.10) [1]:

$$\begin{aligned} \phi_{t+1} &= \phi_t - \frac{\eta}{B} \sum_{i \in \mathcal{I}_t} \nabla_\phi L_i^{(\text{teacher})}(\phi_t) \\ &= \phi_t - \frac{\eta}{B} \sum_{i \in \mathcal{I}_t} \nabla_\phi l_{\text{task}}(g_{\phi_t}(x_i), y_i). \end{aligned} \tag{3}$$

The design of an online teacher is quite demand-oriented, it could be simply a copy of the student network (Li et al.,

---

[1] For clarity, we use the same notation $\eta$, $B$ and $l_{\text{task}}$ to describe the training of $g$ and $f$, although they may select different training algorithms and hyperparameter values in practice.

2022c; Wu & Gong, 2021). But to maintain a valid knowledge gap between student and teacher, they are often initialized using different random seeds in practice. Besides, the training process of teacher network could also be intervened by auxiliary loss from students through reverse distillation (Li & Jin, 2022; Qian et al., 2022; Shi et al., 2021).

**Self KD.** As an alternate approach to a pre-trained teacher, self KD utilizes the hidden information within the student network to guide its learning process. Instead of relying on a large external model, self KD achieves multiple knowledge alignments by introducing auxiliary blocks or creating different representations of the same encoded data. For a block-wise network, $f_\theta = f_{\theta_1} \circ f_{\theta_2} \circ \cdots \circ f_{\theta_m}$ that is composed of $m$ consecutive modules, the whole network $f_\theta$ is regarded as teacher while shallower blocks $f_{\theta_{1 \sim k}} = f_{\theta_1} \circ \cdots \circ f_{\theta_k}$ ($1 \leq k < m$) are students. Following the common setting (Zhang et al., 2019), $\theta$ is updated with multiple task supervision and cross-layer distillation, which in fact can be formulated in terms of $L^{(\text{teacher})}$ in Eq. 3 and $L^{(\text{KD})}$ in Eq. 2 (see Alg. 3 in Appendix A.10):

$$\theta_{t+1} = \theta_t - \frac{\eta}{B} \sum_{i \in \mathcal{I}_t} \nabla_\theta \left[ L_i^{(\text{teacher})}(\theta) + \sum_{k=1}^{m-1} L_i^{(\text{KD})}(\theta_{1 \sim k}, \theta) \right]. \tag{4}$$

## 3.2. Spacing Effect in Biological Learning

Originally discovered by (Ebbinghaus, 2013), the biological spacing effect highlights that the distribution of study sessions across time is critical for memory formation. Then, its functions have been widely demonstrated in various animals and even humans (see Sec. 1). Many cognitive computing models have proposed the concept of spaced learning and described its dynamics, positing an optimal inter-trial interval during memory formation (Landauer, 1969; Peterson, 1966; Wickelgren, 1972). These studies motivate us to further investigate if a proper space interval could benefit KD of possible data variability across training batches. Here we provide more detailed explanations of the interdisciplinary connections:

In machine learning, KD aims to optimize the parameters of a student network with the help of a teacher network by regularizing their outputs to be consistent in response to similar inputs. As shown in a pioneering theoretical analysis (Allen-Zhu & Li, 2020), KD shares a similar mechanism with ensemble learning (EL) in improving generalization from the training set to the test set. In particular, online KD performs this mechanism at temporal scales, and self KD can be seen as a special case of online KD. In comparison, the biological spacing effect can also be generalized to a kind of EL at temporal scales, as the brain network processes similar inputs with a certain time interval and updates its synaptic weights based on previous synaptic weights, which allows for stronger learning performance at test time (Pagani

et al., 2009; Smolen et al., 2016).

The proposed Spaced KD draws inspirations from the biological spacing effect and capitalizes on the underlying connections between KD and EL. It incorporates a space interval between teacher and student to improve generalization. In particular, we hypothesize that an optimal interval may exist between the learning paces of teacher and student in DNNs, as in BNNs.

# 4. Spaced KD

In this section, we describe how Spaced KD is implemented into online KD and self KD, and include a pseudo code for each in Appendix A.10. We then theoretically analyze the benefit of the proposed spacing effect in improving generalization.

## 4.1. Incorporating Spacing Effect into KD

By applying spaced learning in the pipeline of KD, more precisely in the context of online KD, we implement a process of alternate learning between teacher and student. The teacher network updates itself several steps in advance, and then it helps the student network train on the same set of batches. Formally, we define a hyperparameter *Space Interval* denoted as $s$ to represent the gap between the teacher's and student's learning pace. Spaced KD is described as follows (see Fig. 1):

1. First, we train the teacher $g_{\phi_t}(\cdot)$ for $s$ steps (from $\mathcal{B}_t$ to $\mathcal{B}_{t+s-1}$) according to the learning rule in Eq. 3, obtaining an advanced teacher $g_{\phi_{t+s}}(\cdot)$ identical to that of online KD.

2. Then, we freeze the parameters $\phi_{t+s}$ of our teacher $g$, and start to transfer knowledge from it to the student $f_{\theta_t}(\cdot)$ that lags behind over the same batches of training data $\mathcal{B}_{t\sim t+s-1}$:

$$\theta_{t+s} = \theta_t - \frac{\eta}{B} \sum_{j=t}^{t+s-1} \sum_{i \in \mathcal{I}_j} \nabla_\theta L_i^{(\text{KD})}(\theta_j, \phi_{t+s}), \quad (5)$$

where $L_i^{(\text{KD})}$ is the same as Eq. 2 but using fixed teacher parameters $\phi_{t+s}$.

Intrinsically, Spaced KD is a special case of online KD. The main difference that sets Spaced KD apart from online KD is the less frequent updates of the teacher network, which provides a relatively stable learning standard for the student network and potentially contributes to its better generalization ability than the online setting. In practice, we initialize the teacher in Spaced KD using the same random seed as the student. To take a closer look, we theoretically illustrate the impact of the proposed spacing effect on KD with step-by-step mathematical derivations in the next section.

## 4.2. Theoretical Analysis

To understand why Spaced KD might provide better generalization than online KD[2], we analyze the *Hessian matrix* of the loss function for the student network in both scenarios. The Hessian matrix plays a crucial role in understanding the curvature of the loss landscape. In literature, various metrics related to the Hessian matrix have been adopted to evaluate the flatness of a loss minimum after training convergence, reflecting the generalization ability of the trained model (Krizhevsky et al., 2009; Blanc et al., 2020; Damian et al., 2021; Zhou et al., 2020). Here we choose the Hessian trace as a representative for convenience. A smaller Hessian trace indicates a flatter loss landscape, which has also been proved to be related to the upper bound of test set generalization error.

**Setup.** For simplicity we set the dimension of class space as $c = 1$, and the extension of $c > 1$ is straightforward. Let the mean square error (MSE) be the task-specific loss. The KD loss characterizes the distance between two distributions $\hat{y}$ and $y$: $l_{\text{task}}(\hat{y}, y) = l_{\text{KD}}(\hat{y}, y) = \frac{1}{2}(\hat{y} - y)^2$.

**Hessian Matrix.** For KD loss at the $i$-th data sample that follows Eq. 2, the Hessian matrix at a point $\theta$ of student $f_\theta(\cdot)$ with respect to its teacher $g_\phi(\cdot)$ can be calculated as the second-derivative of the empirical risk $L^{(\text{KD})}(\theta, \phi) = \frac{1}{N} \sum_{i=1}^N L_i^{(\text{KD})}(\theta, \phi)$. It could be easily verified that:

$$H_\phi(\theta) = \nabla_\theta^2 L^{(\text{KD})}(\theta, \phi)$$
$$= \frac{1}{N} \sum_{i=1}^N \left[ \nabla_\theta f_\theta(x_i) \nabla_\theta f_\theta(x_i)^\top + \beta(i, \theta, \phi) \nabla_\theta^2 f_\theta(x_i) \right],$$
(6)

where $\beta(i, \theta, \phi) = (1 - \alpha)(f_\theta(x_i) - y_i) + \alpha(f_\theta(x_i) - g_\phi(x_i))$, and in fact $\nabla_\theta L_i^{(\text{KD})}(\theta, \phi) = \beta(i, \theta, \phi) \nabla_\theta f_\theta(x_i)$. At arbitrary time stamp $t$ during the supervised training process, the teacher model's parameters for student $\theta_t$ in online KD is $\phi_t$. In Spaced KD it should be $\phi_{k(t)}$ with $k(t) = (\lceil t/s \rceil)s$ where $\lceil \cdot \rceil$ denotes ceiling operation. Notice that for online KD, the loss function constantly changes due to the update of the teacher, but when we focus on the loss curve for a particular $\phi$, the differentiability of $L_i^{(\text{KD})}$ are preserved, allowing us to continue the discussion.

**Definition 4.1** (Local linearization.)**.** Let $\theta^*$ be a local minimizer of loss function w.r.t $f_\theta(\cdot)$, we call the local linearization of $f_\theta(\cdot)$ at $\theta$ around $\theta^*$ as: $f_\theta(x) = f_{\theta^*}(x) + \langle \theta - \theta^*, \nabla_\theta f_{\theta^*}(x) \rangle$.

For both teacher and student networks, this linearized model in Def. 4.1 provides an applicable approximation of the local dynamic behavior around a converged point. We denote $\phi^*$ and $\theta^*$ as the local minimizer of teacher and student, respectively. Without loss of generality, we assume that

---

[2]For all theoretical analysis and conclusions in this section, we treat self KD as a special case of online KD since they share the same teacher-student relations. In the later Sec. 5, our experiments empirically support this argument as they behave similarly.

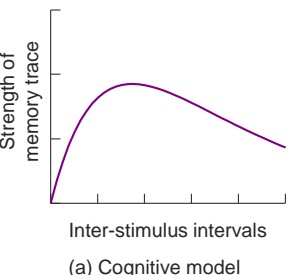

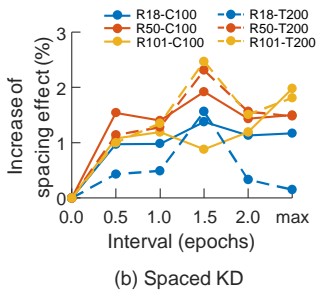

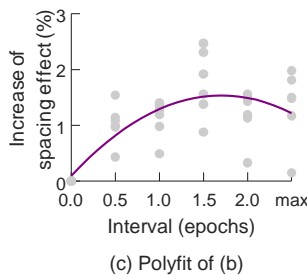

(a) Cognitive model

(b) Spaced KD

(c) Polyfit of (b)

*Figure 2.* Alignment of spaced learning in BNNs and DNNs. **(a)** Computational cognitive model of spaced learning, modified from (Landauer, 1969). **(b)** Overall performance of Spaced KD from different networks and benchmarks. R18: ResNet-18; R50: ResNet-50; R101: ResNet-101; C100: CIFAR-100; T200: Tiny-ImageNet. **(c)** Quadratic polynomial fitting of all performance from **(b)**.

after enough learning steps, $\forall x_i$, $g_{\phi^*}(x_i) = f_{\theta^*}(x_i) = y_i$ which means both models follow the over-parameterized setting so that their training set accuracy eventually become 100%. Therefore, when the student network $f_\theta(\cdot)$ converges to a local minimizer $\theta^*$ in both online KD and Spaced KD, its corresponding teacher network $g_\phi(\cdot)$ should also be close to $\phi^*$ :

$$\beta(i, \theta^*, \phi) = (1-\alpha)(f_{\theta^*}(x_i) - y_i) + \alpha(f_{\theta^*}(x_i) - g_\phi(x_i))$$
$$= \alpha\langle\phi - \phi^*, \nabla_\phi g_{\phi^*}(x_i)\rangle = \alpha\Delta\phi^\top\nabla_\phi g_{\phi^*}(x_i), \quad (7)$$

where $\Delta\phi = \phi - \phi^*$. $\beta$ directly reflects the difference in the teacher model updating between online KD and Spaced KD. We then demonstrate how the combination of mini-batch training and space interval affects the role of the teacher model under the KD framework.

**Definition 4.2** (Teacher model gap). For a teacher model $g_\phi(\cdot)$ trained with SGD using the updating rule in Eq. 3, we define current prediction error over training dataset as the performance gap between $\phi$ and loss minima $\phi^*$: $u(\phi) = \frac{1}{N}\sum_{i=1}^{N}|\Delta\phi^\top\nabla_\phi g_{\phi^*}(x_i)|$.

At a training step $t$ close to convergence (a global time stamp) of the student model, considering the randomness brought by mini-batch sampling, we denote $u_t = \mathbb{E}[u(\phi_t)]$ for online KD, and $u_{k(t)} = \mathbb{E}\left[u\left(\phi_{k(t)}\right)\right]$ for Spaced KD (with space interval $s$) as the parameter gap of their corresponding teacher models, respectively.

**Lemma 4.3** (Lower risk of spaced teacher). $u_{k(t)} \leq u_t$.

*Proof.* It is straightforward that the teacher with $\phi_{k(t)}$ in Spaced KD is an advanced model which has undergone several updating iterations ahead of the student at step $t$. Namely, by definition $t \leq k(t) = (\lceil t/s \rceil)s \leq t + s$. Thus, given the fact that SGD eventually selects a loss minima with linear stability (Wu et al.), i.e., $\mathbb{E}[L^{(\text{teacher})}(\phi_{t+1})] \leq \mathbb{E}[L^{(\text{teacher})}(\phi_t)]$ around $\phi^*$, we have $u_{k(t)} \leq u_t$. $\square$

**Theorem 4.4.** *If the student model $f_\theta(\cdot)$ converges to a local minimizer $\theta^*$ at step $t$ of SGD, let $H_{\phi_t}^{(O)}(\theta^*)$ and $H_{\phi_k}^{(S)}(\theta^*)$ be the Hessian of online KD and Spaced KD, then*

$$\mathbb{E}[Tr(H_{\phi_k}^{(S)}(\theta^*))] \leq \mathbb{E}[Tr(H_{\phi_t}^{(O)}(\theta^*))].$$

The comparison between the Hessian trace for Spaced KD and online KD finally settles in the difference between a spaced but advanced teacher and a frequently updated teacher. Detailed proof of Theorem 4.4 are provided in Appendix A.1 with the help of Lemma 4.3, indicating a flatter loss landscape and thus potentially better generalization ability for the student network of Spaced KD.

**Discussion.** The above analysis reveals key distinctions between Spaced KD, offline KD, and online KD. Spaced KD guides the student $f$ with a well-defined trajectory established by the teacher $g$ that is slightly ahead in training (Shi et al., 2021; Rezagholizadeh et al., 2021), thereby ensuring low errors along such informative direction to improve generalization. With an ideal condition where $g$ and $f$ converge to the same local minima, offline KD and Spaced KD should perform identically best. However, this ideal condition hardly exists in practice, especially given the nature of over-parameterization in advanced DNNs and the complexity of real-world data distributions. These two challenges result in a highly non-convex loss landscape of both $g$ and $f$ with a large number of local minima. Therefore, using a well-trained teacher in offline KD tends to be sub-optimal since $g$ and $f$ can easily converge to different local minima with SGD. In comparison, the limitation of online KD lies in its narrow, constant interval between $g$ and $f$, restricting the exploration of informative directions. By maintaining an appropriate spaced interval, Spaced KD allows for broader explorations and encourages convergence to a more desirable region of the loss landscape, empirically validated in the following section.

# 5. Experiment

In this section, we first describe experimental setups and then present experimental results.

## 5.1. Experimental Setups

**Benchmark.** We evaluate the proposed spacing effect on both ResNet-based architectures (He et al., 2016) such as

ResNet-18, ResNet-50 and ResNet-101, and transformer-based architectures (Dosovitskiy et al., 2020) such as DeiT-Tiny (Touvron et al., 2021) and PiT-Tiny (Heo et al., 2021). We consider four commonly used image classification datasets: CIFAR-100 (Krizhevsky et al., 2009), Tiny-ImageNet, ImageNet-100, and ImageNet-1K (Russakovsky et al., 2015). CIFAR-100 is a well-known image classification dataset of 100 classes and the image size is $32 \times 32$. Tiny-ImageNet consists of 200 classes and the image size is $64 \times 64$. ImageNet-100 and ImageNet-1K contain 100 and 1000 classes of images, respectively, and the image size is $224 \times 224$.

**Implementation.** For ResNet-based architectures, we use an SGD optimizer (Sutskever et al., 2013) with 0.9 momentum, 128 batch size, 80 epochs, and a constant learning rate of 0.01. For KD-related hyperparameters (Zhang et al., 2019), we use a distillation temperature of 3.0, a feature loss coefficient of 0.03, and a KL-Divergence loss weight of 0.3. For transformer-based architectures, we use an AdamW optimizer (Loshchilov & Hutter, 2017a) of batch size 128 and epoch number 300 (warm-up for 20 epochs). Besides, a cosine learning rate decay policy (Loshchilov & Hutter, 2017b) is utilized with initial learning rate $5e-4$ and final $5e-6$, following the training pipeline of previous works (Liu et al., 2021; Li et al., 2022b; Sun et al., 2024).

For Spaced KD, we manually control a sparse interval $s$ in terms of epochs, which is proportional to the total number of samples in the training set (e.g., $s = 0.5$ denotes half of the training set). To avoid potential bias, the training set is shuffled and both teacher and student receive the same data flow. In online KD, the teacher employs the same network architecture as the student if not specified, distilling both response-based (Hinton et al., 2015) and feature-based (Adriana et al., 2015) knowledge. In self KD, the teacher is the deepest layer of the network and the students are the shallow layers along with auxiliary classifiers (Zhang et al., 2019). Specifically, ResNet-based architectures consist of 4 blocks so 3 students correspond to the three shallower blocks. The number of students for transformers depends on the network depth, namely, 11 in our setup. Auxiliary alignment layers and classifier heads are utilized to unify the dimensions of feature and logit vectors produced by students from different depths for distillation. Unless otherwise specified, all results are averaged over three repeats.

### 5.2. Effectiveness and Generality of Spacing Effect

**Overall Performance.** Our proposed Spaced KD outperforms traditional online KD (see Tab. 1) and self KD (see Tab. 2) across different datasets and networks. The performance of different intervals can be seen in Fig. 2 and Tab. 6. Compared to vanilla online KD and self KD, the

*Table 1.* Overall performance of online KD (%). Here are the results for online KD with an interval of 1.5 epochs. The performance of different intervals can be seen in Fig. 2 and Tab. 6. $\Delta$ indicates Spaced KD's performance gain w.r.t online KD.

| Dataset | Network | w/o KD | w/o Ours | w/ Ours | $\Delta$ |
|---|---|---|---|---|---|
| CIFAR-100 | ResNet-18 | 68.12 | 71.05 | **72.43** | +1.38 |
| | ResNet-50 | 69.62 | 71.85 | **73.77** | +1.92 |
| | ResNet-101 | 70.04 | 72.03 | **73.22** | +1.19 |
| | DeiT-Tiny | 64.77 | 65.67 | **67.30** | +1.63 |
| | PiT-Tiny | 73.45 | 74.14 | **75.55** | +1.41 |
| Tiny-ImageNet | ResNet-18 | 53.08 | 59.19 | **60.75** | +1.56 |
| | ResNet-50 | 56.41 | 60.99 | **63.30** | +2.31 |
| | ResNet-101 | 56.99 | 61.29 | **63.76** | +2.47 |
| | DeiT-Tiny | 50.23 | 51.82 | **54.20** | +2.38 |
| | PiT-Tiny | 57.89 | 58.25 | **60.25** | +2.00 |
| ImageNet-100 | ResNet-18 | 77.82 | 78.73 | **80.39** | +1.66 |
| | ResNet-50 | 77.95 | 79.78 | **82.43** | +2.65 |
| | DeiT-Tiny | 70.52 | 70.72 | **73.34** | +2.62 |
| | PiT-Tiny | 76.10 | 76.60 | **78.34** | +1.74 |

*Table 2.* Overall performance of self KD (%). Here are the results for self KD with an interval of 4.0 epochs. $\Delta$ indicates Spaced KD's performance gain w.r.t self KD.

| Dataset | Network | w/o KD | w/o Ours | w/ Ours | $\Delta$ |
|---|---|---|---|---|---|
| CIFAR-100 | ResNet-18 | 68.12 | 73.29 | **75.73** | +2.44 |
| | ResNet-50 | 69.62 | 75.73 | **78.73** | +3.00 |
| | ResNet-101 | 70.04 | 76.16 | **79.24** | +3.08 |
| | Deit-Tiny | 64.77 | 65.24 | **68.26** | +3.02 |
| Tiny-ImageNet | ResNet-18 | 53.08 | 61.08 | **62.83** | +1.75 |
| | ResNet-50 | 56.41 | 63.58 | **65.80** | +2.22 |
| | ResNet-101 | 56.99 | 63.35 | **66.79** | +3.44 |
| | Deit-Tiny | 50.17 | 49.73 | **53.59** | +3.86 |
| ImageNet-100 | ResNet-18 | 77.82 | 76.21 | **79.27** | +3.06 |
| | Deit-Tiny | 69.52 | 70.50 | **73.46** | +2.96 |

enhancement of accuracy is **2.14%** on average, with moderate variations from a minimum of 1.19% on ResNet-101 / CIFAR-100 to a maximum of 3.44% on ResNet-101 / Tiny-ImageNet. For the larger dataset ImageNet-1K, our Spaced KD improves the performance for ResNet-18 and ViT networks by up to 5.08% (see Tab. 7, Tab. 8 of Appendix A.4).

**Teacher-Student Gap.** Considering that capacity gaps between teacher and student for their different architectures or training progress would affect distillation gains (see Sec. 2), we further evaluate various teacher-student pairs across model sizes and architectures, and Spaced KD remains effective in all cases (see Tab. 9 and Tab. 10 in Appendix A.5). Interestingly, if we train the teacher ahead of the student by $s$ steps at the beginning and then distill its knowledge to the student maintaining a constant training gap, there is no significant improvement over the online KD (see Tab. 11). This indicates the particular strength of Spaced KD, which applies in the later stage rather than the early stage.

*Table 3.* Performance of Spaced KD on ResNet-18 / CIFAR-100 using different loss functions.

| Loss Function | w/o | w/ $s = 1.5$ | $\Delta$ |
|---|---|---|---|
| L1 | **69.54** | 69.30 | -0.24 |
| Smooth L1 | 68.96 | **69.45** | +0.49 |
| MSE (Mean) | 69.34 | **69.45** | +0.11 |
| MSE (Sum) | 71.05 | **72.43** | +1.38 |
| Cross-Entropy | 70.38 | **72.04** | +1.66 |

*Table 4.* Performance of Spaced KD on ResNet-18 / CIFAR-100 using more recent KD methods.

| Dataset/Model | Method | TSB | CTKD | LSKD |
|---|---|---|---|---|
| CIFAR-100/ResNet-18 | w/o KD | 67.65 | 67.86 | 67.94 |
| | w/ KD | 71.70 | 69.41 | 70.74 |
| | w/ Ours | **72.82** | **71.12** | **71.76** |
| CIFAR-100/DeiT-Tiny | w/o KD | 51.90 | 53.31 | 52.31 |
| | w/ KD | 52.63 | 54.20 | 52.93 |
| | w/ Ours | **55.47** | **54.72** | **53.83** |
| Tiny-ImageNet/ResNet-18 | w/o KD | 55.21 | 53.03 | 54.05 |
| | w/ KD | 59.92 | 58.78 | 59.30 |
| | w/ Ours | **61.65** | **60.32** | **60.28** |
| Tiny-ImageNet/DeiT-Tiny | w/o KD | 40.29 | 40.82 | 39.65 |
| | w/ KD | 40.13 | 41.22 | 41.14 |
| | w/ Ours | **43.36** | **41.60** | **41.48** |

**Different KD Losses.** To evaluate generality, we implement Spaced KD with representative loss functions, such as L1, smooth L1, MSE (reduction=mean), MSE (reduction=sum), and cross-entropy. As shown in Tab. 3, Spaced KD applies to different loss functions with consistent improvements.

**Different KD Methods.** We combine our Spaced KD with other more advanced KD methods, including (1) TSB (Li et al., 2022a) constructs superior "teachers" with temporal accumulator and spatial integrator; (2) CTKD (Li et al., 2023) controls the task difficulty level during the student's learning career through a dynamic and learnable temperature; (3) LSKD (Sun et al., 2024) employs a plug-and-play Z-score pre-process of logit standardization before applying softmax and KL divergence. As shown in Tab. 4, Spaced KD brings significant improvements to a wide range of KD methods. The above results suggest that the benefits of Spaced KD arise from the fundamental properties of parameter optimization in deep learning, consistent with our theoretical analysis in Sec. 4.4.

### 5.3. Extended Analysis of Spacing Effect

**Sensitivity of Space Interval.** Through extensive investigation (see Fig. 2 and Tab. 6 in Appendix A.2), the space interval $s$ is relatively insensitive and $s = 1.5$ results in consistently strong improvements. Therefore, we selected it as the default choice to obtain the performance of our Spaced

KD in all comparisons. This property also largely avoids the computational cost and complexity of model optimization imposed by the new hyperparameter.

**Critical Period of Spaced KD.** In order to better understand the underlying mechanisms of Spaced KD, we empirically investigate the critical period of implementing the proposed spacing effect. As shown in Fig. 3, we control the start time of spaced distillation throughout the training process, and discover that initiating Spaced KD in the later stage of training is more beneficial than the early stage for performance improvements of the student network. This suggests that in KD, not only the interval between learning sessions but also the timing of spaced learning are important. Unlike previous understandings that attribute the KD efficacy to the knowledge capacity gap between the teacher and the student (where Spaced KD should be more effective in the early stage of training, see Sec. 2), our results point out a novel direction for KD research from a temporal perspective. Specifically, the "right time to learn" is critical for the student, and the teacher could influence the student's convergence to a better solution by intervening during the later training stage.

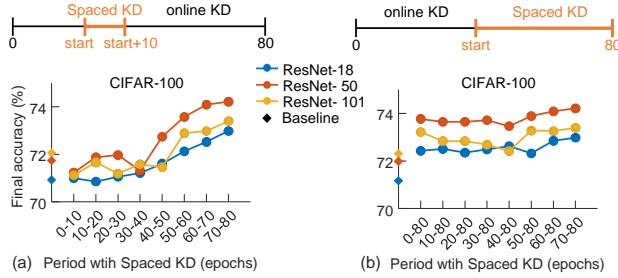

*Figure 3.* Impact of different initiating times of Spaced KD ($s = 1.5$), which is introduced **(a)** for constant 10 training epochs or **(b)** till the end of training.

**Learning Rate and Batch Size.** As described in previous works, the learning rate and batch size influence the endpoint curvature and the whole trajectory (Frankle et al.; Lewkowycz et al., 2020; Xie et al., 2020). The learning rate corresponds to the parameters' updating step length, and batch size would affect the total number of updating iterations which directly relates to the choice of space interval $s$. Therefore, we further validate the impact of learning rate and batch size. As shown in Fig. 5 of Appendix A.7, we summarize the results: (i) Spaced KD proves effective w.r.t naive online KD and self KD across different learning rates; (ii) Spaced KD exhibits its advantages when training with a relatively large batch size (greater than 64). These observations also align with previous research (Jastrzebski et al., 2019; Wu et al.) regarding a small batch size limiting

the maximum spectral norm along the convergence path found by SGD from the beginning of training.

## 5.4. Generalization of Spaced KD

**Flat Minima.** To verify whether Spaced KD could converge to a flat minima, we conduct experiments to observe the model robustness that reflects the flatness of loss landscape around convergence, following previous works (Zhang et al., 2018; 2019). We first train ResNet-18/50/101 networks on CIFAR-100 with traditional online KD (`w/o`) and our Spaced KD (`w/1.5`, the interval is 1.5 epochs). Then Gaussian noise is added to the parameters of those models to evaluate their training loss and accuracy over the training set at various perturbation levels, which are plotted in Fig. 4. The results show that the model trained with Spaced KD maintains a higher accuracy and lower loss deviations than naive KD under gradient noise level. Furthermore, after applying this interference, the training loss of the independent model significantly increases, whereas the loss of the Spaced KD model rises much less. These results suggest that the model with Spaced KD has found a much wider minima, which is likely to result in better generalization performance.

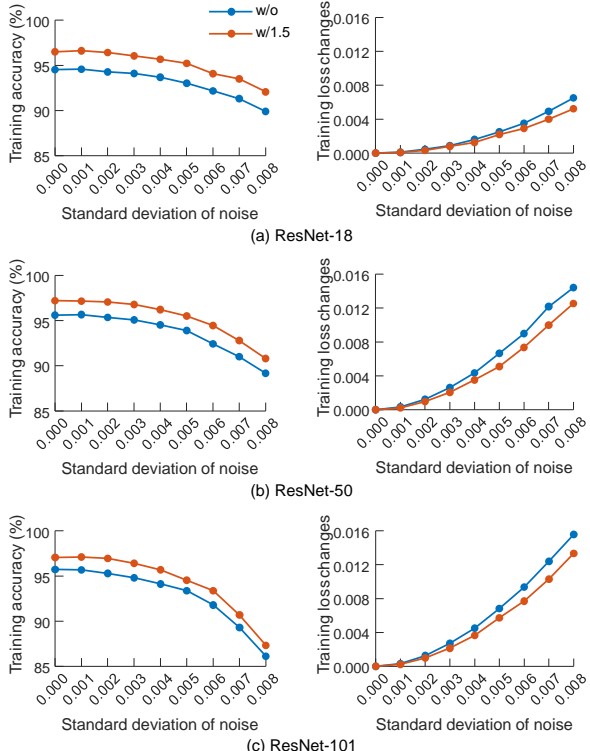

*Figure 4.* Impact of Gaussian noise on performance.

**Noise Robustness.** In addition to manipulating network parameters, we conduct an extra experiment to evaluate the

model's generalization ability to multiple transformations that create out-of-distribution images. Specifically, we apply 6 representative operations of image corruption (Michaelis et al., 2019) to the images of the CIFAR-100 test set. The test accuracy at noise intensity 1.0 is recorded in Tab. 5 and results of other intensity levels can be found in Tab. 12 of Appendix A.8. It is clear that in most cases with different corruption types and network architectures, our proposed Spaced KD helps the student network resist noise attacks, which reflects its superior robustness to unseen inference situations. Besides, we test robust accuracy using a representative adversarial attack method called BIM (Kurakin et al., 2017), and our Spaced KD is more robust across different architectures (see Tab. 13 in Appendix A.9). The above results empirically offer evidence for the generalization promotion brought by the spacing effect.

## 6. Conclusion

In this paper, we present Spaced Knowledge Distillation (Spaced KD), a bio-inspired strategy that is simple yet effective for improving online KD and self KD. We theoretically demonstrate that the spaced teacher helps the student model converge to flatter local minima via SGD, resulting in better generalization. With extensive experiments, Spaced KD achieves significant performance gains across a variety of benchmark datasets, network architectures and baseline methods, providing innovative insights into the learning paradigm of KD from a temporal perspective. Since we also reveal a possible critical period of spacing effect and provide its potential theoretical implications in DNNs, our findings may offer computational inspirations for neuroscience. By exploring more effective spaced learning paradigms and investigating detailed neural mechanisms, our work is expected to facilitate a deeper understanding of both biological learning and machine learning.

Although our approach has achieved remarkable improvements, it also has potential *limitations*: Our results suggest a relatively insensitive optimal interval ($s = 1.5$) for Spaced KD, yet remain under-explored its theoretical foundation and an adaptive strategy for determining it. Additionally, our results indicate that the timing of Spaced KD is important. The effectiveness of adaptive adjusting the space interval and the timing of distillation remains to be validated and analyzed in subsequent research. In future work, we would actively explore the application of such spacing effect for a broader range of scenarios, such as curriculum learning, continual learning, and reinforcement learning.

## Acknowledgements

This work was supported by the STI2030-Major Projects (2022ZD0204900 to Y.Z.), the NSFC Projects

*Table 5.* Comparison of accuracy under image corruption attack (%). Δ indicates Spaced KD's performance gain w.r.t online KD. The intensity of noise is 1.0 and the results of other intensities (i.e., 3.0, 5.0) can be seen in Tab. 12 of Appendix. A.8.

| Method | Attack | ResNet-18 | | | ResNet-50 | | | ResNet-101 | | |
|---|---|---|---|---|---|---|---|---|---|---|
| | | w/o Ours | w/ Ours | Δ | w/o Ours | w/ Ours | Δ | w/o Ours | w/ Ours | Δ |
| Online KD | impulse_noise | 52.92 | **54.15** | 1.23 | 55.41 | **57.19** | 1.78 | 55.66 | **57.17** | 1.51 |
| | zoom_blur | 66.45 | **67.43** | 0.98 | 66.86 | **68.53** | 1.67 | 66.20 | **66.53** | 0.33 |
| | snow | 57.28 | **59.05** | 1.77 | 59.19 | **59.55** | 0.36 | 57.82 | **58.38** | 0.56 |
| | frost | 57.55 | **59.64** | 2.09 | **60.16** | 59.93 | -0.23 | 58.57 | **59.63** | 1.06 |
| | jpeg_compression | **41.12** | 40.23 | -0.89 | 42.22 | **42.65** | 0.43 | 42.69 | **43.70** | 1.01 |
| | brightness | 67.01 | **69.06** | 2.05 | 68.31 | **69.15** | 0.84 | 66.82 | **67.45** | 0.63 |
| Self KD | impulse_noise | 50.65 | **60.57** | 9.92 | 62.18 | **71.57** | 9.39 | 59.33 | **68.78** | 9.45 |
| | zoom_blur | 64.44 | **68.60** | 4.16 | 68.03 | **72.13** | 4.10 | 66.09 | **71.16** | 5.07 |
| | snow | 61.30 | **66.14** | 4.84 | 64.72 | **69.65** | 4.93 | 64.03 | **68.76** | 4.73 |
| | frost | 63.96 | **67.80** | 3.84 | 66.36 | **71.81** | 5.45 | 66.73 | **70.21** | 3.48 |
| | jpeg_compression | 30.99 | **34.67** | 3.68 | 34.44 | **35.34** | 0.90 | 33.64 | **34.76** | 1.12 |
| | brightness | 73.18 | **75.92** | 2.74 | 74.91 | **79.19** | 4.28 | 75.10 | **78.91** | 3.81 |

(Nos. 62406160 to L.W., 32021002 to Y.Z.), and National Science and Technology Major Project (2022ZD01163013 to B.L.). L.W. is also supported by the Postdoctoral Fellowship Program of CPSF under Grant Number GZB20230350 and the Shuimu Tsinghua Scholar.

## Impact Statement

This paper presents work whose goal is to advance the field of Machine Learning. There are many potential societal consequences of our work, none which we feel must be specifically highlighted here.

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

# A. Appendix

## A.1. Proof of Theorem 4.4

*Proof.* For a general KD loss, we have the trace of its Hessian matrix at global minimizer $\theta^*$:

$$
\begin{aligned}
\mathrm{Tr}\left(H_\phi(\theta^*)\right) &= \frac{1}{N}\sum_{i=1}^{N}\left[\|\nabla_\theta f_{\theta^*}(x_i)\|^2 + \beta(i,\theta^*,\phi)\mathrm{Tr}\left(\nabla_\theta^2 f_{\theta^*}(x_i)\right)\right] \\
&= \frac{1}{N}\sum_{i=1}^{N}\left[\|\nabla_\theta f_{\theta^*}(x_i)\|^2 + \alpha\Delta\phi^\top\nabla_\phi g_{\phi^*}(x_i)\mathrm{Tr}\left(\nabla_\theta^2 f_{\theta^*}(x_i)\right)\right].
\end{aligned}
\tag{8}
$$

For online KD and Spaced KD, the expectation of their Hessian trace should be:

$$
\mathbb{E}[\mathrm{Tr}(H_{\phi_t}^{(O)}(\theta^*))] = \mathbb{E}_i[\|\nabla_\theta f_{\theta^*}(x_i)\|^2] + \frac{\alpha}{N}\sum_{i=1}^{N}\mathbb{E}\left[\Delta\phi_t^\top\nabla_\phi g_{\phi^*}(x_i)\mathrm{Tr}\left(\nabla_\theta^2 f_{\theta^*}(x_i)\right)\right],
\tag{9}
$$

$$
\mathbb{E}[\mathrm{Tr}(H_{\phi_k}^{(S)}(\theta^*))] = \mathbb{E}_i[\|\nabla_\theta f_{\theta^*}(x_i)\|^2] + \frac{\alpha}{N}\sum_{i=1}^{N}\mathbb{E}\left[\Delta\phi_k^\top\nabla_\phi g_{\phi^*}(x_i)\mathrm{Tr}\left(\nabla_\theta^2 f_{\theta^*}(x_i)\right)\right].
\tag{10}
$$

By Lemma 4.3, $\frac{1}{N}\sum_i\mathbb{E}[|\Delta\phi_k^\top\nabla_\phi g_{\phi^*}(x_i)|] \leq \frac{1}{N}\sum_i\mathbb{E}[|\Delta\phi_t^\top\nabla_\phi g_{\phi^*}(x_i)|]$,

$$
\frac{\alpha}{N}\sum_{i=1}^{N}\mathbb{E}\left[\Delta\phi_k^\top\nabla_\phi g_{\phi^*}(x_i)\mathrm{Tr}\left(\nabla_\theta^2 f_{\theta^*}(x_i)\right)\right] \leq \frac{\alpha}{N}\sum_{i=1}^{N}\mathbb{E}\left[\Delta\phi_t^\top\nabla_\phi g_{\phi^*}(x_i)\mathrm{Tr}\left(\nabla_\theta^2 f_{\theta^*}(x_i)\right)\right].
$$

Substituting the above inequality into Eq. 9 and Eq. 10 completes the proof. □

## A.2. Performance of Different Intervals for Online KD

*Table 6.* Overall performance of different intervals for Fig. 2 and Tab. 1.

| Dataset | Network | Baseline | w/o | w/0.5 | w/1.0 | w/1.5 | w/2.0 | w/max |
|---|---|---|---|---|---|---|---|---|
| | ResNet-18 | 68.12 | 71.05 | 72.02 | 72.03 | **72.43** | 72.18 | 72.22 |
| | ResNet-50 | 69.62 | 71.85 | 73.39 | 73.25 | **73.77** | 73.28 | 73.35 |
| CIFAR-100 | ResNet-101 | 70.04 | 72.03 | 73.11 | 73.22 | 72.91 | 73.22 | **74.01** |
| | DeiT-Tiny | 64.77 | 65.67 | 66.03 | 66.22 | **67.30** | 66.45 | 65.69 |
| | PiT-Tiny | 73.45 | 74.14 | **75.55** | 75.50 | 75.27 | 75.12 | 74.07 |
| | ResNet-18 | 53.08 | 59.19 | 59.62 | 59.68 | **60.75** | 59.52 | 59.34 |
| | ResNet-50 | 56.41 | 60.99 | 62.13 | 62.27 | **63.30** | 62.55 | 62.47 |
| Tiny-ImageNet | ResNet-101 | 56.99 | 61.29 | 62.70 | 62.64 | **63.76** | 62.80 | 63.10 |
| | DeiT-Tiny | 50.23 | 51.82 | **54.20** | 53.55 | 52.92 | 53.48 | 52.21 |
| | PiT-Tiny | 57.89 | 58.25 | 59.45 | 59.77 | **60.25** | 59.75 | 58.23 |

## A.3. Implementation of SOTA methods with Spaced KD

For traditional KD methods (BAN (Furlanello et al., 2018), TAKD (Mirzadeh et al., 2020a)) and online KD methods (DML (Zhang et al., 2018) and SHAKE (Li & Jin, 2022)), we preserve their basic training frameworks for reproducing results in w/o KD (raw ResNet-18 training) and KD (ResNet-18 with the corresponding method) columns and delay the students' supervised learning and distillation by a space interval of 1.5 epochs for w/ Ours. For self KD methods (DLB (Shen et al., 2022) and PSKD (Kim et al., 2021)), we initiate a student network identical to the teacher. We train the teacher model utilizing PSKD or DLB, and the student model is trained either online or in a spaced style with an interval of

1.5 epochs. Specifically, the results `w/o KD` of PSKD and DLB in Tab. 4 are the performance of the teacher model, `w/ KD` is the performance of online students, and `w/ Ours` corresponds to spaced students. Because we follow the exact training pipeline (including learning rate scheduler, optimizer, and dataset transformation, etc) of those works when reproducing their results, which is different from that of Tab. 1 and Tab. 2, the baselines without KD may be different.

## A.4. Performance of Spaced KD on ImageNet-1k

*Table 7.* Performance of ResNet-18 on ImageNet-1k Dataset (space interval 1.5 epochs).

| Epoch (epoch) | 20 | 40 | 60 | 80 |
|---|---|---|---|---|
| online KD | 44.35 | 45.94 | 46.62 | 48.59 |
| online KD w/ Ours | 49.28 | 51.35 | 52.36 | 53.67 |
| self KD | 44.87 | 46.78 | 47.78 | 47.99 |
| self KD w/ Ours | 47.03 | 49.94 | 50.22 | 51.57 |

*Table 8.* Performance of Deit-Tiny on ImageNet-1k Dataset (space interval 1.5 epochs).

| Epoch (epoch) | 100 | 200 | 300 |
|---|---|---|---|
| online KD | 58.18 | 65.93 | 72.04 |
| online KD w/ Ours | 58.47 | 66.54 | 72.34 |
| self KD | 58.81 | 66.37 | 72.39 |
| self KD w/ Ours | 60.82 | 67.27 | 73.69 |

## A.5. Performance of Spaced KD on Different Teacher-Student Architectures

*Table 9.* Overall performance of student networks distilled from different teachers on CIFAR-100. We use ResNet-18 as the student.

| | Teacher | Baseline | Online KD | Spaced KD |
|---|---|---|---|---|
| | ResNet-18×2 | 69.40 | 71.77 | **72.77** |
| Width | ResNet-18×4 | 70.75 | 72.17 | **73.11** |
| | ResNet-18×8 | 70.77 | 72.03 | **73.52** |
| Depth | ResNet-50 | 69.21 | 72.18 | **73.49** |
| | ResNet-101 | 69.54 | 71.61 | **73.04** |
| Architecture | DeiT-Tiny | 64.65 | 78.61 | **79.38** |
| | PiT-Tiny | 73.78 | 77.13 | **78.77** |

*Table 10.* Comparison of Spaced KD and offline KD from different teacher-student pairs on CIFAR-100. We use ResNet-18 as the student.

| | Teacher | Offline KD | Spaced KD |
|---|---|---|---|
| | ResNet-18×2 | 72.53 | **72.77** |
| Size | ResNet-18×4 | 72.83 | **73.11** |
| | ResNet-18×8 | 73.04 | **73.52** |
| Architecture | DeiT-Tiny | 78.80 | **79.38** |
| | Pit-Tiny | 78.50 | **78.77** |

### A.6. Performance of Student Distilled from a Constant Ahead Teacher

*Table 11.* Performance of ResNet-18 on CIFAR-100 distilled from trained teacher with a constant $s$ step ahead. There are no significant improvements over the online KD.

| Interval (epoch) | 0 | 0.5 | 1 | 1.5 | 2 | 2.5 |
|---|---|---|---|---|---|---|
| CIFAR-100 | 71.05 | 70.67 | 70.85 | 70.69 | 71.04 | 70.78 |

Here we consider a naive baseline of implementing the proposed spacing effect. Specifically, we first train the teacher model for $s$ steps and then transfers knowledge to the student model at each step during the following training time. In other words, the teacher model keeps constant $s$ steps ahead of the student model. However, such a naive baseline exhibits no significant improvement over online KD (see Table 11), consistent with our empirical analysis (see Fig. 3) and theoretical analysis (see Sec. 4.2): The teacher model of Spaced KD can provide a stable informative direction for optimizing the student model after each $s$ steps, whereas the teacher model of the naive baseline fails in this purpose due to its ongoing changes when optimizing the student model. Such different effects also suggest that the implementation of spacing effect is highly non-trivial and requires specialized design as in our Spaced KD.

### A.7. Performance of Spaced KD using different learning rate and batch size

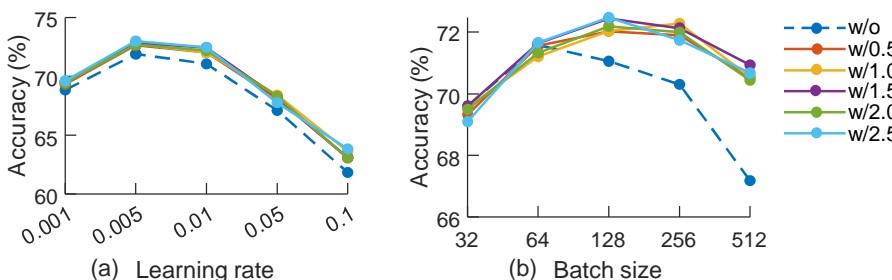

(a)  Learning rate         (b)  Batch size

*Figure 5.* Hyperparameter validation for Spaced KD. Accuracy of different learning rate **(a)** and batch size **(b)** of gradient intervals.

### A.8. Performance of Spaced KD on Different Image Corruption Attacks

Here we visualize 6 representative image corruption operations (Michaelis et al., 2019) applied to the images from the CIFAR-100 dataset (Krizhevsky et al., 2009) to assess our models' robustness and generalization ability in Fig. 6. The accuracy under adversarial attacks with more noise intensity levels is listed in Tab. 12.

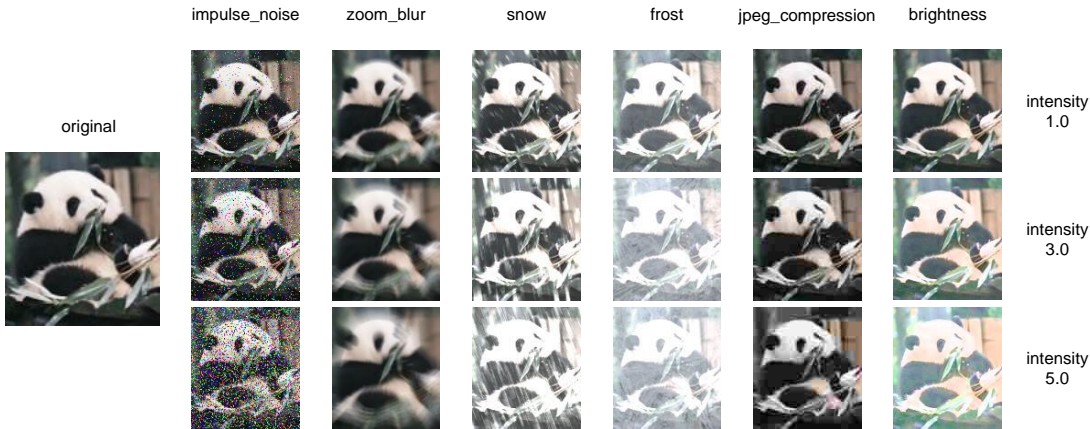

*Figure 6.* Image corruption operation. We choose 6 representative image corruption operations with different severity (1.0, 3.0, 5.0) and visualized images come from the CIFAR-100 test set.

## A.9. Performance of Spaced KD after Adversarial Attack

## A.10. Pseudo Code of Online KD, Self KD and Spaced KD

---
**Algorithm 1** Training Algorithm of Online KD
---
**Require**: student $f_\theta$, teacher $g_\phi$, dataset $\mathcal{D}_{\text{train}}$, KD loss weight $\alpha$, epoch number $E$
**Ensure**: train both teacher and student using online knowledge distillation

1: **for** $1 \le e \le E$ **do**
2:  **for** $(x_i, y_i) \in \mathcal{D}_{\text{train}}$ **do**
3:    Update teacher $\phi \leftarrow \phi - \nabla_\phi l_{\text{task}}(g_\phi(x_i), y_i)$
4:    Update student $\theta \leftarrow \theta - \nabla_\theta \left[\alpha l_{\text{KD}}(f_\theta(x_i), g_\phi(x_i)) + (1 - \alpha)l_{\text{task}}(f_\theta(x_i), y_i)\right]$
5:  **end for**
6: **end for**
---

---

**Algorithm 2** Training Algorithm of Online KD with Spaced KD

---

**Require**: student $f_\theta$, teacher $g_\phi$, dataset $\mathcal{D}_{\text{train}}$, KD loss weight $\alpha$, epoch number $E$, space interval $s$
**Ensure**: train both teacher and student using spaced knowledge distillation

1: Initialize data index set: $\mathcal{R} \leftarrow \emptyset$
2: **for** $1 \leq e \leq E$ **do**
3:     **for** $(x_i, y_i) \in \mathcal{D}_{\text{train}}$ **do**
4:         $\mathcal{R} \leftarrow \mathcal{R} \cup \{i\}$
5:         Update teacher $\phi \leftarrow \phi - \nabla_\phi l_{\text{task}}(g_\phi(x_i), y_i)$
6:         **if** $|\mathcal{R}| == s$ **then**
7:             **for** $j \in \mathcal{R}$ **do**
8:                 Retrieve $(x_j, y_j)$ from $\mathcal{D}_{\text{train}}$
9:                 Update student $\theta \leftarrow \theta - \nabla_\theta \left[ \alpha l_{\text{KD}}(f_\theta(x_j), g_\phi(x_j)) + (1-\alpha) l_{\text{task}}(f_\theta(x_j), y_j) \right]$
10:            **end for**
11:         **end if**
12:         Clear index set: $\mathcal{R} \leftarrow \emptyset$
13:     **end for**
14: **end for**

---

---

**Algorithm 3** Training Algorithm of Self KD

---

**Require**: network $f_\theta = f_{\theta_1} \circ \cdots \circ f_{\theta_m}$ consisting of $m$ blocks, dataset $\mathcal{D}_{\text{train}}$, KD loss weight $\alpha$, epoch number $E$
**Ensure**: train $f_\theta$ by distilling logits from the last block to the shallower blocks

1: **for** $1 \leq e \leq E$ **do**
2:     **for** $(x_i, y_i) \in \mathcal{D}_{\text{train}}$ **do**
3:         Calculate loss $L = l_{\text{task}}(f_\theta(x_i), y_i)$
4:         **for** $1 \leq k < m$ **do**
5:             $L \leftarrow L + \alpha l_{\text{KD}}(f_{\theta_1} \circ \cdots \circ f_{\theta_k}(x_i), f_\theta(x_i)) + (1-\alpha) l_{\text{task}}(f_{\theta_1} \circ \cdots \circ f_{\theta_k}(x_i), y_i)$
6:         **end for**
7:         Update network $\theta \leftarrow \theta - \nabla_\theta L$
8:     **end for**
9: **end for**

---

*Table 12.* Comparison of accuracy under image corruption attack (%). $\Delta$ indicates Spaced KD's increased performance based on online KD. The results of 1.0 intensity can be seen in Tab. 5.

| Attack | Noise Intensity | ResNet-18 | | | ResNet-50 | | | ResNet-101 | | |
|---|---|---|---|---|---|---|---|---|---|---|
| | | w/o Ours | w/ Ours | $\Delta$ | w/o Ours | w/ Ours | $\Delta$ | w/o Ours | w/ Ours | $\Delta$ |
| impulse_noise | 3.0 | 34.19 | **35.33** | 1.14 | 35.41 | **36.53** | 1.12 | 37.56 | **38.16** | 0.60 |
| | 5.0 | **12.54** | 12.04 | -0.50 | 10.49 | **10.57** | 0.08 | **12.08** | 11.39 | -0.69 |
| zoom_blur | 3.0 | 64.73 | **65.29** | 0.56 | 65.04 | **66.45** | 1.41 | 64.5 | **64.98** | 0.48 |
| | 5.0 | 61.02 | **61.53** | 0.51 | 61.36 | **62.67** | 1.31 | 61.32 | **62.18** | 0.86 |
| snow | 3.0 | 44.48 | **45.42** | 0.94 | 46.91 | **47.17** | 0.26 | 44.5 | **45.87** | 1.37 |
| | 5.0 | 28.60 | **29.48** | 0.88 | **30.09** | 29.71 | -0.38 | 30.09 | **30.75** | 0.66 |
| frost | 3.0 | 42.40 | **43.10** | 0.70 | **44.87** | 44.69 | -0.18 | 45.10 | **45.28** | 0.18 |
| | 5.0 | 37.80 | **39.47** | 1.67 | 39.26 | **39.97** | 0.71 | **41.24** | 40.59 | -0.65 |
| jpeg_compression | 3.0 | **33.23** | 32.32 | -0.91 | 33.05 | **33.99** | 0.94 | 34.80 | **35.63** | 0.83 |
| | 5.0 | 20.75 | **21.32** | 0.57 | 20.29 | **20.86** | 0.57 | 21.55 | **22.29** | 0.74 |
| brightness | 3.0 | 62.77 | **64.68** | 1.91 | 64.48 | **64.63** | 0.15 | 62.90 | **64.01** | 1.11 |
| | 5.0 | 54.11 | **54.56** | 0.45 | 55.34 | **55.46** | 0.12 | 54.47 | **55.71** | 1.24 |

*Table 13.* Performance of Spaced KD on CIFAR-100 after an adversarial attack called BIM (Kurakin et al., 2017). Spaced KD is more robust than online KD.

| Network | ResNet-18 | ResNet-50 | ResNet-101 |
|---------|-----------|-----------|------------|
| w/o | 31.33 | 31.32 | 31.70 |
| w/1.5 | **31.44** | **31.70** | **33.69** |
| $\Delta$ | +0.11 | +0.38 | +1.99 |

---

**Algorithm 4** Training Algorithm of Self KD with Spaced KD

---

**Require**: network $f_\theta = f_{\theta_1} \circ \cdots \circ f_{\theta_m}$ consisting of $m$ blocks, dataset $\mathcal{D}_{\text{train}}$, KD loss weight $\alpha$, epoch number $E$, space interval $s$

**Ensure**: train $f_\theta$ by distilling logits from the last block to shallower blocks in a spaced manner

1: Initialize data index set: $\mathcal{R} \leftarrow \emptyset$
2: **for** $1 \le e \le E$ **do**
3:     **for** $(x_i, y_i) \in \mathcal{D}_{\text{train}}$ **do**
4:         $\mathcal{R} \leftarrow \mathcal{R} \cup \{i\}$
5:         Calculate loss $L = l_{\text{task}}(f_\theta(x_i), y_i)$
6:         Update network $\theta \leftarrow \theta - \nabla_\theta L$
7:     **if** $|\mathcal{R}| == s$ **then**
8:         **for** $j \in \mathcal{R}$ **do**
9:             Retrieve $(x_j, y_j)$ from $\mathcal{D}_{\text{train}}$
10:            Calculate loss $L' = l_{\text{task}}(f_\theta(x_j), y_j)$
11:            **for** $1 \le k < m$ **do**
12:                $L' \leftarrow L' + \alpha l_{\text{KD}}(f_{\theta_1} \circ \cdots \circ f_{\theta_k}(x_j), f_\theta(x_j)) + (1 - \alpha) l_{\text{task}}(f_{\theta_1} \circ \cdots \circ f_{\theta_k}(x_j), y_j)$
13:            **end for**
14:            Update network $\theta \leftarrow \theta - \nabla_\theta L'$
15:         **end for**
16:     **end if**
17:     Clear index set: $\mathcal{R} \leftarrow \emptyset$
18:     **end for**
19: **end for**

---

