# OpenReview forum: "Right Time to Learn: Promoting Generalization via Bio-inspired Spacing Effect in Knowledge Distillation"
_ICML.cc/2025/Conference — ICML 2025 poster_

### Official Review · Reviewer_rhLU · 2025-02-19

**Overall Recommendation:** 3

**Summary:**

This paper, inspired by the spacing effect, proposes Spaced KD, which distills the student model by a teacher who pretrains s-steps ahead of the student. This paper demonstrates theoretically that Spaced KD produces flatter loss landscapes, and proves experimentally the superior performance of Spaced KD in both online KD and self KD scenarios.

## update after rebuttal

Further responses from the authors show that their method leads to a slight increase in training time without other additional overhead, and can be combined with other KD methods to improve performance.

**Claims And Evidence:**

The authors claim that Spaced KD has a lower update frequency of the teacher model compared to online KD. However, Spaced KD updates the teacher model s steps ahead of the student model, as shown in Figure 1 and Algorithm 2, and updates the teacher model s times per advance, which does not reduce the number of updates to the teacher model. I would like the author to explain this.

**Essential References Not Discussed:**

This paper presents preliminaries for understanding its method and cites relevant papers in Section 3. However, I would still recommend including citations and comparisons to more recent KD literature (years 2023 & 2024).

**Experimental Designs Or Analyses:**

1. I suggest that the authors include the additional overhead that Spaced KD incurs compared to vanilla KD, such as memory and time during training.
2. It is recommended that the authors provide the accuracy of the teacher model in their experiments to show the performance gap between Spaced KD trained students and teachers. In other words, is it possible that Spaced KD-trained students can outperform teachers? As far as I know, some distillation methods have been able to enable their students to perform beyond the teacher model.
3. In Table 5, the authors should provide a comparison with other distillation methods.
4. The authors achieved further performance gains by using Spaced KD on other KD methods in Table 4. However, I consider these methods to be too dated and suggest that the authors use the most recent KD methods as a baseline to further demonstrate the generalizability of their method.

**Methods And Evaluation Criteria:**

Spaced KD is easy to understand and implement, and experimental results show that Spaced KD further improves distillation performance in both online and self KD.

However, I am still confused about how Spaced KD applies to self KD. [1] uses the final output to guide the intermediate output for self-distillation, so the teacher and student are the same model but with different capacities. This self-distillation is designed to reduce distillation overhead. If Spaced KD is used in self-distillation, is it necessary to additionally store the parameters of the teacher model, which is updated s steps ahead during training? How much would this increase the training overhead?

[1] Be your own teacher: Improve the performance of convolutional neural networks via self distillation. In ICCV. 2019.

**Other Comments Or Suggestions:**

It is recommended that the authors elaborate further on the details and overhead of Spaced KD.

**Other Strengths And Weaknesses:**

I believe that the method proposed in this paper can contribute to the development of bio-inspired KD algorithms, and the experimental results demonstrate the superior performance of Spaced KD. However, I am concerned about the overhead of Spaced KD during training, which, while easy to implement, has the potential to incur significant additional memory/time overhead, thus affecting distillation efficiency, especially for self-distillation with the goal of efficient distillation.

**Questions For Authors:**

I have no further questions.

**Relation To Broader Scientific Literature:**

The method proposed in this paper is simple and easy to implement, and can be better combined with previous distillation methods to further improve performance.

**Theoretical Claims:**

There is no obvious error in the theoretical analysis presented in this paper.

---

> ### Author Rebuttal · Authors · 2025-03-31
>
> Thank you for your valuable comments. We provide a point-to-point response as follows. We hope you may consider this a sufficient reason to raise the score. If you have any further questions, please let us know.
>
> **Q1: The additional overhead that Spaced KD incurs compared to vanilla KD.**
>
> The computational and parameter overhead of our Spaced KD is essentially **identical**  the counterparts of online KD and self KD. Regardless of whether or not Spaced KD is introduced, the entire model (both teacher and student) employs the same network architecture and is trained with the same total number of epochs. In practice, since we need to train the teacher s epochs in advance and wait for the student to follow, this results in a slight delay in the runtime (around 30%) but does **not** increase the computational overhead (i.e., the waiting teacher is frozen, not computing).
>
> **Q2: Lower update frequency of the teacher model.**
>
> Our claim "less frequent" means that, the teacher is distilled every s epochs in Spaced KD, rather than every iteration in online KD. We will clarify this in the final version.
>
> **Q3: How Spaced KD applies to self KD and the additional parameter/training overhead.**
>
> In self KD, the deepest layer (as the teacher) transfers knowledge to the shallow layers (as the student) at each training time step [1]. In its spaced version, we first train the model using the cross-entropy loss between the deepest layer's output and the ground-truth label for $s$ steps. We then train the model using the standard self KD loss [1] between the deepest layer's output and each shallow layer's output for $s$ steps. Therefore, our spaced version of self KD does **not** store additional parameters and does **not** increase the training overhead (i.e., the total number of epochs is not changed). We will make this clearer.
>
> [1] Be your own teacher: Improve the performance of convolutional neural networks via self distillation. ICCV, 2019.
>
> **Q4: The accuracy of the teacher model and its comparison to the student model.**
>
> We would point out that in all experiments, "w/o KD" denotes the teacher's performance, "w/o Ours" and "w/ KD" denote the student's performance after online KD or self KD without our spacing effect, while "w/ Ours" denotes the student's performance after online KD or self KD with our spacing effect (referred to as the Spaced KD).
> This is because existing online KD and self KD methods often employ (two copies of) the same network as both teacher and student to improve generalization of the model itself. Therefore, the student's performance will exceed the teacher's performance if the online KD and self KD methods work well.
>
> **Q5: In Table 5, the authors should provide a comparison with other distillation methods.**
>
> In addition to online KD in Table 5, we add more experiments of **self KD** with different corruption types and network architectures. As shown in the following table, Spaced KD can also largely improve the robustness of self KD in generalizing to different noisy scenarios.
>
> |Attack|ResNet18 w/o Ours|ResNet18 w/ Ours|ResNet50 w/o Ours|ResNet50 w/ Ours|ResNet101 w/o Ours|ResNet101 w/ Ours|
> |-|-|-|-|-|-|-|
> |impulse_noise|50.65|**60.57**|62.18|**71.57**|59.33|**68.78**|
> |zoom_bhr|64.44|**68.60**|68.03|**72.13**|66.09|**71.16**|
> |snow|61.30|**66.14**|64.72|**69.65**|64.03|**68.76**|
> |frost|63.96|**67.80**|66.36|**71.81**|66.73|**70.21**|
> |jpeg_compression|30.99|**34.67**|34.44|**35.34**|33.64|**34.76**|
> |brightness|73.18|**75.92**|74.91|**79.19**|75.10|**78.91**|
>
> **Q6 & Q7: Include citations and comparisons to more recent KD methods.**
>
> Following your suggestion, we have included more recent KD methods, especially those for 2023-2024. For example, TSB [1] constructs superior "teachers" with temporal accumulator and spatial integrator. CTKD [2] controls the task difficulty level during the student’s learning career through a dynamic and learnable temperature. LSKD [3] employs a plug-and-play Z-score pre-process of logit standardization before applying softmax and KL divergence.
>
> As shown in the following table, our Spaced KD can be combined with these methods and provide significant improvements under two datasets and two architectures.
>
> |Dataset/Model|Method|TSB [1]|CTKD [2]|LSKD [3]|
> |-|-|-|-|-|
> |CIFAR-100/ResNet-18|w/o KD|67.65|67.86|67.94|
> | |w/ KD|71.70|69.41|70.74|
> | |w/ Ours|**72.82**|**71.12**|**71.76**|
> |CIFAR-100/DeiT-Tiny|w/o KD|51.90|53.31|52.31|
> | |w/ KD|52.63|54.20|52.93|
> | |w/ Ours|**55.47**|**54.72**|**53.83**|
> |Tiny-IN/ResNet-18|w/o KD|55.21|53.03|54.05|
> | |w/ KD|59.92|58.78|59.30|
> | |w/ Ours|**61.65**|**60.32**|**60.28**|
> | Tiny-IN/DeiT-Tiny|w/o KD|40.29|40.82|39.65|
> | |w/ KD|40.13|41.22|41.14|
> | |w/ Ours|**43.36**|**41.60**|**41.48**|
>
> [1] Online Knowledge Distillation by Temporal-Spatial Boosting. WACV, 2022. (requested by Reviewer C5Ts)
> [2] Curriculum Temperature for Knowledge Distillation. AAAI, 2023.
> [3] Logit Standardization in Knowledge Distillation. CVPR, 2024.

---

> > ### Comment · Reviewer_rhLU · 2025-04-05
> >
> > Thanks to the authors' response, additional results suggest that Spaced KD can indeed improve the performance of some recent KD methods. However, I still have a few questions. However, I still have some concerns that I would like the author to address. As Reviewer bqdc suggests, the authors need to compare with more SOTA KD methods. Also, the author may have misunderstood my initial comment. It is hoped that the authors will compare their method to the SOTA KD methods propsed in other papers in Table 5, rather than just to their baseline, so that the performance advantages of their method can be further confirmed.

---

> > > ### Author Response · Authors · 2025-04-08
> > >
> > > Thank you for you timely feedback. Since our Spaced KD is designed to improve generalization of a deep learning model itself with the same architecture and dataset, we mainly consider online KD, self KD, and relatively advanced methods of these two avenues in our paper (as well as our rebuttal results). It is worth noting that many state-of-the-art KD methods are not intended for this purpose (most of them focus on distilling a large teacher to a small student), making it difficult to adapt between our and their settings.
> > >
> > > We have tried our best to search online / self KD methods and find that DTSKD [1] is the state-of-the-art, whose reported results outperform MixSKD (ECCV'22), PS-KD (CVPR'21), Tf-KD (CVPR'20), CS-KD (CVPR'20), etc. We run its officially released code and implement Spaced KD on it. For CIFAR-100 with VGG-16, ResNet-18 and ResNext-18, the improved performance of Spaced KD over DTSKD (running the default setting of its officially released code) is **+1.40%**, **+0.88%** and **+1.24%** under exactly the same configuration, which demonstrates the advantages of our approach.
> > > Due to the very limited time in rebuttal, we will finish the experiments of other architecture / dataset setups and add them in the final version.
> > >
> > > [1] Dual teachers for self-knowledge distillation. Pattern Recognition, 2024.

---

### Official Review · Reviewer_C5Ts · 2025-03-13

**Overall Recommendation:** 3

**Summary:**

The paper introduces Spaced Knowledge Distillation (Spaced KD), a novel method drawing on the biological spacing effect to enhance generalization in online and self-knowledge distillation by inserting intervals between teacher and student training steps. It makes notable contributions: firstly, the bio-inspired strategy of adding temporal intervals between teacher updates and student distillation, which results in flatter loss landscapes ; secondly, the theoretical analysis that shows Spaced KD converges to flatter minima through Hessian trace analysis; and lastly, the empirical validation that reveals significant performance improvements.

**Claims And Evidence:**

Performance improvements (Tables 1–2) are backed by extensive experiments.
Flat minima hypothesis is validated via noise robustness tests (Fig. 4, Page 8) and Hessian analysis (Sec. 4.2).

**Essential References Not Discussed:**

Online Knowledge Distillation by Temporal-Spatial Boosting
C. Li, Z. Wang and H. Qi, "Online Knowledge Distillation by Temporal-Spatial Boosting," doi: 10.1109/WACV51458.2022.00354.
A recent work on rehearsal-based KD with temporal intervals, which shares conceptual similarities but is not cited.

**Experimental Designs Or Analyses:**

Strengths: Ablation studies on interval sensitivity (Fig. 2, Page 6) and critical timing (Fig. 3, Page 7) are thorough.
Weakness: The baseline comparison lacks state-of-the-art methods (e.g., contrastive distillation). Tables 1–2 focus on older baselines (e.g., BAN, DML).

**Methods And Evaluation Criteria:**

Methods: Spaced KD introduces a temporal interval  s  (e.g., 1.5 epochs) between teacher updates and student distillation, compatible with existing KD frameworks (Algorithms 1–3, Appendix).
Evaluation: Standard benchmarks (CIFAR-100, Tiny-ImageNet) and metrics (test accuracy) are appropriate. However, ImageNet-1K results are less comprehensive (Table 7, Page 14).  The architectures validated in the experiments mainly include variants of ResNet, DeiT and PiT. Including more diverse architectures, such as WRN and VGG, would help to verify the effectiveness of the method. Additionally, It is mentioned( Page 2, Line 69 ) that it has a plug - in effect for a wide range of self - distillation and online distillation methods. However, the improvements over existing methods (Table 4) do not follow the latest methods, and the validated architectures/datasets are only ResNet - 18/CIFAR - 100.

**Other Comments Or Suggestions:**

Clarity: The term "space interval" (Page 4) could be confused with spatial intervals; "temporal interval" is more precise.

**Other Strengths And Weaknesses:**

Originality: Novel integration of neuroscience principles into KD.
Clarity: Well-structured, but the pseudo-code (Appendix A.10) lacks implementation details (e.g., gradient accumulation for interval s).

**Questions For Authors:**

Why is the optimal interval s = 1.5 epochs (Page 7)? Is this dataset-dependent? A theoretical justification is missing.

**Relation To Broader Scientific Literature:**

Connects KD with biological spacing effect (Page 1, Lines 30–40), leveraging prior work on flat minima (Keskar et al., 2016) and online/self-KD (Zhang et al., 2018).

**Theoretical Claims:**

Theorem 4.4 (Page 4) links Spaced KD to flatter minima via Hessian trace analysis. The proof (Appendix A.1) assumes over-parameterized models and linearized dynamics, which may not fully capture real-world DNN training.

---

> ### Author Rebuttal · Authors · 2025-03-31
>
> Thank you for your valuable comments. We provide a point-to-point response as follows. We hope you may consider this a sufficient reason to raise the score. If you have any further questions, please let us know.
>
> **Q1: ImageNet-1K results are less comprehensive in Table 7.**
>
> We would respectfully point out that the ImageNet-1K results are presented in both Table 7 and Table 8, including different KD paradigms (online KD and self KD) and network architectures (ResNet-18 and Deit-Tiny). Our Spaced KD provides improvements in all cases.
>
> **Q2: Evaluation of more recent KD methods with more scenarios.**
>
> As shown in the response to **Reviewer bqdc's Q1**, our proposed Spaced KD brings significant improvements to a range of more recent KD methods under two datasets and two architectures. We will add these results in the final version.
>
> **Q3: The proof (Appendix A.1) assumes over-parameterized models and linearized dynamics, which may not fully capture real-world DNN training.**
>
> We would respectfully argue that state-of-the-art DNNs are often **over-parameterized** to improve generalization. This property results in multiple local minima with similar training errors but different testing errors, often reflected in the flatness of loss landscape. We therefore adopt the over-parameterization assumption to analyze how to converge to a flatter loss landscape.
>
> Also, we would point out that we assume **local linearization** around the convergence point, rather than global linearization. As shown in previous work [1], SGD eventually selects a loss minima with linear stability (i.e., low errors with moderate linear disturbance), therefore ensuring flat loss landscape and improving generalization.
> We will add more explanations to these two assumptions in the final version.
>
> [1] The alignment property of SGD noise and how it helps select flat minima: A stability analysis. NeurIPS, 2022.
>
> **Q4: Some details (e.g., hyperparameters for transformer training) are missing.**
>
> For transformer training, we adopt the well-established training pipeline tailored for the benchmark datasets [1-3]. We will add more detailed descriptions in the final version.
>
> [1] Efficient Training of Visual Transformers with Small Datasets, NeurIPS. 2021.
> [2] Locality Guidance for Improving Vision Transformers on Tiny Datasets. ECCV, 2022.
> [3] Logit Standardization in Knowledge Distillation. CVPR, 2024.
>
> **Q5: One related work [1] is not cited.**
>
> Thanks. We have conceptually analyzed this work and empirically demonstrated Spaced KD's plug-in benefit on it (see response to **Reviewer bqdc's Q1**).
>
> **Q6: The pseudo-code (Appendix A.10) lacks implementation details (e.g., gradient accumulation for interval s).**
>
> We would clarify that our proposed Spaced KD inherently avoids gradient accumulation through its dual-loop design. The hyperparameter s (i.e., temporal interval) operates as a temporal decoupler rather than a gradient accumulation window. With the outer loop, the teacher model updates its parameters immediately after each batch (Algorithm 2, line 5), with no gradient retention. With the inner loop (when $|\mathcal{R}=s$), the teacher’s parameters remain frozen while the student updates using the cached samples, ensuring no backward passes occur through the teacher in this phase. This design strictly segregates the gradient flows: the teacher’s gradients are computed and applied instantly in the outer loop, while the student’s gradients are confined to the inner loop without cross-interval persistence. We will add more explanations to make this clearer.
>
> **Q7: The "temporal interval" is more precise than "space interval".**
>
> Following your suggestion, we will modify our expression for better clarity.
>
> **Q8: Why is the optimal interval s = 1.5 epochs (Page 7)? Is this dataset-dependent? A theoretical justification is missing.**
>
> We theoretically demonstrate that there exists a desirable temporal interval to improve generalization of online KD (as well as self KD) and empirically investigate the selection of specific values (see online KD in Table 6, and self KD in the response to **Reviewer bqdc's Q2**). We find that the desirable temporal interval and its effectiveness is related to the strength of SGD-induced variability (target of KD paradigms, learning rate, batchsize, etc.), and is relatively **insensitive** to different network architectures and benchmark datasets.
>
> Since the SGD-induced variability of online KD (targeting the entire network) is larger than that of self KD (targeting a few network blocks), the former requires a smaller temporal interval to aggregate such variability to obtain an appropriate teacher-student gap. We therefore select s=1.5 for online KD and s=4.0 for self KD as the default implementation. We further investigate the impact of learning rate and batch size in Fig.5, which together with the temporal interval affect the overall performance.

---

> > ### Comment · Reviewer_C5Ts · 2025-04-08
> >
> > Thanks for the careful revision. The experimental analysis can get enhanced in the formal version.

---

> > > ### Author Response · Authors · 2025-04-08
> > >
> > > Thank you for your timely feedback. We will definitely include the additional experiments and the enhanced analysis in the formal version.

---

### Official Review · Reviewer_bqdc · 2025-03-13

**Overall Recommendation:** 2

**Summary:**

This paper proposes a space KD strategy, which is inspired by spacing effect in biological learning and memory. Overall, the experiments verified the effectiveness when comparing the proposed space KD with self KD(Zhang et al., 2019). However, there are many KD methods been proposed in recent years, it missing comparisons with SOTA KD methods.

**Claims And Evidence:**

Overall the claims are well supported. However, whether the space KD is compatible with other SOTA KD methods is not well analyzed.

**Essential References Not Discussed:**

Missing review on latest KD methods.

**Experimental Designs Or Analyses:**

1. Missing comparisons with SOTA KD methods

2. Missing experiments on whether the propose strategy is compatible with SOTA KD methods.

**Methods And Evaluation Criteria:**

The evaluation criteria is reasonable

**Other Comments Or Suggestions:**

See [Other Strengths And Weaknesses]

**Other Strengths And Weaknesses:**

1. Missing comparisons with SOTA KD methods

2. Missing experiments on whether the propose strategy is compatible with SOTA KD methods.

3. Missing review on latest KD methods.

4. In online KD, the knowledge distillation performance will be affected by the capacity gap. The discussion in 'Teacher-Student Gap' is shallow. It is not clear when and why the Spaced KD is not effective in online KD.

5. Missing discussions on the selection of Intervals for self KD and online KD.

6. Missing failure analysis. It is not clear when the spaced KD not works.

**Questions For Authors:**

No

**Relation To Broader Scientific Literature:**

No problems with this part.

**Theoretical Claims:**

The theoretical basis is the  spacing effect in biological learning and memory. I think it is correct.

---

> ### Author Rebuttal · Authors · 2025-03-31
>
> Thank you for your valuable comments. We provide a point-to-point response as follows. We hope you may consider this a sufficient reason to raise the score. If you have any further questions, please let us know.
>
> **Q1: Review, comparison, and compatibility experiment on latest KD methods.**
>
> Following your suggestion, we have included more recent KD methods. Here we discuss some representative ones: TSB [1] constructs superior "teachers" with temporal accumulator and spatial integrator. CTKD [2] controls the task difficulty level during the student’s learning career through a dynamic and learnable temperature. LSKD [3] employs a plug-and-play Z-score pre-process of logit standardization before applying softmax and KL divergence.
>
> As shown in the following table, our Spaced KD can be combined with these methods and provide significant improvements under two datasets and two architectures.
>
> |Dataset/Model|Method|TSB [1]|CTKD [2]|LSKD [3]|
> |-|-|-|-|-|
> |CIFAR-100/ResNet-18|w/o KD|67.65|67.86|67.94|
> | |w/ KD|71.70|69.41|70.74|
> | |w/ Ours|**72.82**|**71.12**|**71.76**|
> |CIFAR-100/DeiT-Tiny|w/o KD|51.90|53.31|52.31|
> | |w/ KD|52.63|54.20|52.93|
> | |w/ Ours|**55.47**|**54.72**|**53.83**|
> |Tiny-IN/ResNet-18|w/o KD|55.21|53.03|54.05|
> | |w/ KD|59.92|58.78|59.30|
> | |w/ Ours|**61.65**|**60.32**|**60.28**|
> | Tiny-IN/DeiT-Tiny|w/o KD|40.29|40.82|39.65|
> | |w/ KD|40.13|41.22|41.14|
> | |w/ Ours|**43.36**|**41.60**|**41.48**|
>
> [1] Online Knowledge Distillation by Temporal-Spatial Boosting. WACV, 2022.
> [2] Curriculum Temperature for Knowledge Distillation. AAAI, 2023.
> [3] Logit Standardization in Knowledge Distillation. CVPR, 2024.
>
> **Q2: Selection of intervals for self KD and online KD.**
>
> We theoretically demonstrate that there exists a desirable temporal interval to improve generalization of online KD (as well as self KD) and empirically investigate the selection of specific values (see online KD in Table 6, and self KD in the following table). We find that the desirable temporal interval for a given KD paradigm is relatively stable across different network architectures and benchmark datasets.
> Since the SGD-induced variability of online KD (targeting the entire network) is larger than that of self KD (targeting a few network blocks), the latter requires a larger temporal interval to aggregate such variability to obtain an appropriate teacher-student gap. We therefore select s=1.5 for online KD and s=4.0 for self KD as the default implementation.
>
> Temporal interval of self KD:
> |Interval (epochs)|0|1|2|3|4|8|
> |-|-|-|-|-|-|-|
> |CIFAR-100/ResNet-18|73.25|74.27|75.15|74.30|75.73|76.41|
> |CIFAR-100/ResNet-50|75.73|76.67| 79.27|79.89|79.43|79.44|
> |CIFAR-100/ResNet-101|76.16|75.97| 79.02|79.27|79.24|79.64|
>
> **Q3 & Q4: More discussion of ``Teacher-Student Gap'' in online KD and its failure analysis.**
>
> We agree that the capacity gap between teacher and student is critical to the online KD performance. In previous studies of (online) KD, the teacher-student gap is regulated in various dimensions, such as the training data, network architecture, innate randomness (random initialization and SGD), etc. In this work, our main motivation lies in improving generalization of the model itself with **identical** network architectures, random initialization and data sources between teacher and student. We therefore characterize the teacher-student gap in online KD into the interplay of the SGD-induced variability and the temporal interval that aggregates such variability.
>
> Theoretically, we demonstrate that a proper temporal interval between teacher and student (i.e., a proper teacher-student gap) helps the model to find flatter local minima using SGD. This is because the teacher that is slightly ahead in training provides a well-defined trajectory, ensuring low errors along so-called informative direction to improve generalization. In contrast, the naive SGD only ensures low errors in random directions around the convergence point, which limits the "radius" of loss flatness (see Sec.4.2).
>
> Empirically, we perform extensive experiments to validate our proposal in online KD across a range of datasets and architectures. Aligned with our theoretical analysis, the proposed Spaced KD is most effective only when the temporal interval is set to an appropriate value (Fig.2 and Table 6). When the temporal interval is too large, the spaced version of online KD is closer to offline KD and the improvement is compromised. This is because the too large teacher-student gap makes them converge to different local minima, which violates the assumption of our theoretical analysis.
>
> In the absence of differences in other spatial elements that produce the teacher-student gap, the effectiveness of our Spaced KD is regulated by the temporal interval in SGD. We believe that the effectiveness of our Spaced KD will be affected if the teacher-student gap produced by such spatial elements is too large, which is out of scope of the current focus.

---

### Decision · Program_Chairs · 2025-05-01

**Decision:**

Accept (poster)

**Comment:**

This paper proposes Spaced Knowledge Distillation inspired by the "spacing effect" in biological learning, where spaced learning intervals improve learning and generalization. The authors provide theoretical analysis suggesting spaced knowledge distillation helps DNNs find flatter loss minima during training, which is linked to better generalization. Experiments across multiple benchmarks and architectures demonstrate that the proposed method yields notable performance gains without additional training costs.

Reviewers generally acknowledged the originality of integrating spacing effect into knowledge distillation and found the theoretical motivation, which connects the spacing effect to achieving flatter loss minima for better generalization, plausible. Initial experimental results were also recognized for demonstrating performance improvements over standard online distillation and self-distillation baselines. Despite the appreciated novelty, a recurring concern raised by the reviewers was the missing comparison with broader SOTA in knowledge distillation; reviewers pointed out missing comparisons, compatibility tests, and literature review concerning more recent and advanced KD methods beyond the basic online/self KD paradigms initially compared against. Another area of concern involved the practical overhead of Spaced KD, with specific questions about potential increases in memory usage and training time, particularly questioning its efficiency within self KD.

The authors provided a comprehensive rebuttal and presented new experimental results demonstrating that Spaced KD is compatible with and provides significant performance boosts when integrated with several recent SOTA KD methods (including TSB, CTKD, LSKD, and DTSKD). They clarified that the method incurs minimal additional computational or memory overhead, explaining the slight runtime delay and detailing its application to self KD without extra parameter storage. Additionally, they offered justifications for the interval selection process based on SGD variability, elaborated on their theoretical framework and assumptions, and clarified technical points like the meaning of update frequency.

This rebuttal successfully addressed the primary concerns for reviewers who engaged with it, leading to positive final assessments and recommendations leaning towards acceptance, although some still noted a desire for broader SOTA comparisons. In terms of review scores, the paper is borderline. Nevertheless, weighing the novel neuroscience connection and the strong rebuttal, while also noting the varying reviewer engagement post-rebuttal, I believe this submission can be a useful addition to the conference and recommend acceptance.